# The Ratio of von Willebrand Factor Antigen to ADAMTS13 Activity: Usefulness as a Prognostic Biomarker in Acute-on-Chronic Liver Failure

**DOI:** 10.3390/biology12020164

**Published:** 2023-01-20

**Authors:** Hiroaki Takaya, Tadashi Namisaki, Masahide Enomoto, Takahiro Kubo, Yuki Tsuji, Yukihisa Fujinaga, Norihisa Nishimura, Kosuke Kaji, Hideto Kawaratani, Kei Moriya, Takemi Akahane, Masanori Matsumoto, Hitoshi Yoshiji

**Affiliations:** 1Department of Gastroenterology, Nara Medical University, Kashihara 634-8522, Japan; 2Department of Blood Transfusion Medicine, Nara Medical University, Kashihara 634-8522, Japan

**Keywords:** ADAMTS13, von Willebrand factor, biomarkers, liver cirrhosis, acute-on-chronic liver failure

## Abstract

**Simple Summary:**

ACLF has a high risk of short-term mortality. ADAMTS13:AC and VWF:Ag are associated with ACLF development. We investigated the relationship between VWF:Ag/ADAMTS13:AC and prognosis of ACLF. In total, 101 patients with cirrhosis were enrolled in this study. The VWF:Ag/ADAMTS13:AC was associated with prognosis in the patients with ACLF in multivariate analysis. The cumulative survival of the patients with ACLF was significantly lower for patients with high VWF:Ag/ADAMTS13:AC compared with those with low VWF:Ag/ADAMTS13:AC. The VWF:Ag/ADAMTS13:AC predicted prognosis in patients with cirrhosis with ACLF.

**Abstract:**

Acute-on-chronic liver failure (ACLF) has a high risk of short-term mortality. A disintegrin-like and metalloproteinase with thrombospondin type-1 motifs 13 (ADAMTS13) is a metalloproteinase that specifically cleaves multimeric von Willebrand factor (VWF). Imbalance between ADAMTS13 and VWF is associated with portal hypertension, which induces ACLF development. A previous study reported that ADAMTS13 activity (ADAMTS13:AC) and VWF antigen (VWF:Ag) are predictive biomarkers of ACLF development in patients with cirrhosis. This study investigated the changes in ADAMTS13:AC and VWF:Ag levels from before to after the development of ACLF to determine their usefulness as a prognostic biomarker in patients with ACLF. In total, 101 patients with cirrhosis were enrolled in this study. The level of ADAMTS13:AC and VWF:Ag was determined by an enzyme-linked immunosorbent assay. Cox proportional hazard regression analysis was conducted to determine independent prognostic factors for patients with liver cirrhosis in the post-ACLF group. ADAMTS13:AC levels gradually decreased in the order of non-ACLF group, pre-ACLF group, and finally post-ACLF group. VWF:Ag and the ratio of VWF:Ag to ADAMTS13:AC (VWF:Ag/ADAMTS13:AC) levels gradually increased in the order of non-ACLF group, pre-ACLF group, followed by post-ACLF group. VWF:Ag/ADAMTS13:AC and CLIF-C ACLF scores were associated with prognosis in the post-ACLF group in multivariate analysis. The cumulative survival of the post-ACLF group was significantly lower for patients with high VWF:Ag/ADAMTS13:AC (>9) compared with those with low VWF:Ag/ADAMTS13:AC (≤9) (HR: 10.72, 95% confidence interval: 1.39–82.78, *p* < 0.05). The VWF:Ag/ADAMTS13:AC increased according to the progression of ACLF in patients with cirrhosis and predicted prognosis in patients with cirrhosis with ACLF.

## 1. Introduction

A disintegrin-like and metalloproteinase with thrombospondin type-1 motifs 13 (ADAMTS13) is a metalloproteinase that specifically cleaves multimeric von Willebrand factor (VWF) between the Tyr1605 and Met1606 residues in the A2 domain [1,2,3,4]. ADAMTS13 is produced exclusively in hepatic stellate cells adjacent to endothelial cells (ECs) [5]. VWF is synthesized in vascular ECs and released into the plasma as unusually large VWF multimers [6]. In the state of imbalance of the ADAMTS13 enzyme–VWF substrate, VWFs are improperly cleaved, causing them to accumulate and to induce platelet thrombus formation in the microvasculature under high shear stress [7]. The imbalance in the ADAMTS13 enzyme–VWF substrate (i.e., the decrease in ADAMTS13 activity (ADAMTS13:AC) and increase in VWF antigen (VWF:Ag)) was associated with the severity of liver cirrhosis (LC) [2,8,9], acute liver failure (ALF) [3], and severe alcoholic hepatitis [10,11]. Our previous study reported that the imbalance of the ADAMTS13 enzyme–VWF substrate worsened according to LC progression [2,8,9,12], and that patients with LC with hepatic encephalopathy, ascites, and hepatorenal syndrome (HRS) had a more significant imbalance in the ADAMTS13 enzyme–VWF substrate than those without these manifestations [2,8,9]. As a result, patients with LC with a significant imbalance in the ADAMTS13 enzyme–VWF substrate had a higher risk of mortality than those without this imbalance [2,13,14], i.e., ADAMTS13:AC and VWF:Ag have become useful prognostic biomarkers for patients with LC [2,13,14]. Furthermore, patients with ALF and severe alcoholic hepatitis with a significant imbalance in the ADAMTS13 enzyme–VWF substrate had a higher mortality risk than those without the imbalance as well as LC [3,10,11,15].

Acute-on-chronic liver failure (ACLF) develops in patients with LC after a bacterial infection, gastrointestinal bleeding, alcohol intake, or worsening of the underlying liver disease [16,17]. ACLF has a high risk of short-term mortality associated with the development of multiple organ failure [18]. A previous study reported that portal hypertension (PHT) induces the development of ACLF [13,18], because PHT triggers infection (e.g., spontaneous bacterial peritonitis) and gastrointestinal bleeding (e.g., esophagogastric variceal bleeding) [19,20], which are important triggers of ACLF [16,17]. The imbalance in the ADAMTS13 enzyme–VWF substrate worsened according to the severity of PHT [21,22,23]. A previous study reported that VWF:Ag has become a useful diagnostic biomarker of PHT [23,24]; moreover, our previous study revealed that the ratio of VWF:Ag to ADAMTS13:AC (VWF:Ag/ADAMTS13:AC) has become a useful predictive biomarker of the development of ACLF [13]. It is difficult to prevent ACLF development in patients with LC because it is impossible to reverse a cirrhotic liver state to a normal liver state. Therefore, more accurate prognostic markers in patients with ACLF are required. VWF:Ag/ADAMTS13:AC may become a prognostic biomarker as well as a predictive biomarker in patients with ACLF.

This study investigated the changes in the imbalance of the ADAMTS13 enzyme–VWF substrate from before to after the development of ACLF, to determine whether ADAMTS13:AC and VWF:Ag can be used as a prognostic biomarker in patients with ACLF.

## 2. Methods

### 2.1. Patients and Study Design

This retrospective observational study included a series of 101 patients with LC whose ADAMTS13:AC and VWF:Ag levels were assayed in our hospital between August 2012 and October 2021 (Figure 1). Among these patients, 34 developed ACLF (these patients were included in the post-ACLF group) but 67 did not, at the time of the assay of ADAMTS13:AC and VWF:Ag; 34 patients with ACLF did not have venous thromboembolism (VTE). Samples were collected on admission for ACLF. A total of 11 of 34 patients with ACLF had infections (4 had cellulitis, 3 had sepsis, 2 had spontaneous bacterial peritonitis, 1 had appendicitis, and 1 had pyelonephritis). Sixty-seven patients without ACLF did not have VTE or infection. Samples were collected during hospital visits for LC. Among the 34 patients with ACLF, 21 patients survived, whereas 13 died during the observation period. The median observation period was 16 days (interquartile range 8–29). Among the 67 patients without ACLF, 54 patients did not develop ACLF during the observation period (these patients were included in the non-ACLF group), whereas 13 developed ACLF during the observation period (these patients were included in the pre-ACLF group). The median observation period was 1386 (interquartile range 888–1494) days. LC was diagnosed based on physical findings, laboratory tests, and imaging according to the 2020 evidence-based clinical practice guidelines for LC from the Japan Society of Gastroenterology and the Japan Society of Hepatology [25,26]. ACLF was diagnosed based on laboratory tests according to the diagnostic criteria for acute-on-chronic liver failure and related disease conditions in Japan [18]. These diagnostic criteria included patients with LC with Child–Pugh class A and B that had prothrombin time (PT) ≤ 40% and total bilirubin (T-Bil) ≥ 5.0 mg/dL. Patients with incurable hepatocellular carcinoma were excluded according to these diagnostic criteria. All the patients in the non-ACLF group and pre-ACLF group had Child–Pugh class A and B but did not have incurable hepatocellular carcinoma, and all the patients in the post-ACLF group did not have incurable hepatocellular carcinoma in the present study. We initially investigated the relationship of ADAMTS13:AC and VWF:Ag with the non-ACLF group, pre-ACLF group, or post-ACLF group in patients with LC. Subsequently, we studied the relationship of ADAMTS13:AC and VWF:Ag with prognosis of patients with LC in the post-ACLF group. This study was approved by the local ethics committee of Nara Medical University and was performed in accordance with the ethical standards of the Declaration of Helsinki. Informed consent was obtained from all participants included in the study.

### 2.2. Determination of ADAMTS13 Activity and VWF Antigen

The samples were stored in plastic tubes containing 0.38% sodium citrate. Platelet-poor plasma was prepared by centrifuging at 3000× *g* at 4 °C for 15 min and stored in aliquots at −80 °C until analysis. Plasma ADAMTS13:AC was determined via the sensitive chromogenic enzyme-linked immunosorbent assay (ELISA) (Kainos Laboratories Inc., Tokyo, Japan) [27]. The normal values of ADAMTS13:AC were 99% ± 22%. Plasma VWF:Ag was measured by sandwich ELISA using a rabbit antihuman VWF polyclonal antiserum (Dako, Glostrup, Denmark). The normal values of VWF:Ag were 102% ± 33% [28].

### 2.3. Statistical Analysis

The statistical analysis was performed using EZR (Saitama Medical Center, Jichi Medical University), which is a graphical user interface for R (version 4.1.2, R Foundation for Statistical Computing, https://www.r-project.org, last accessed on 10 January 2023). EZR is a modified version of the R commander version 2.7–1 that includes statistical functions frequently used in biostatistics [29]. The results were reported as medians and interquartile ranges. The differences between the groups were first analyzed using the Kruskal–Wallis rank test, followed by the Steel–Dwass test in groups showing significant differences. Categorical data were analyzed using Fisher’s exact test. Cox proportional hazard regression analysis was conducted to determine independent prognostic factors for patients with liver cirrhosis in the post-ACLF group. A two-tailed *p*-value of <0.05 was considered significant.

## 3. Results

### 3.1. Clinical Characteristics of the Patients

The patient characteristics are shown in Table 1. The median age of patients with LC was 71 (62–75) years. The study population comprised 64 men and 37 women. Five patients had hepatitis B virus, 23 patients had hepatitis C virus, 40 patients had a condition of alcohol abuse, 14 patients had nonalcoholic steatohepatitis, 7 patients had autoimmune hepatitis, 5 patients had primary biliary cholangitis, and 7 patients had other conditions. The albumin (Alb) level decreased gradually according to the progression of ACLF. The T-Bil and creatinine (Cre) levels increased gradually with the progression of ACLF. The aspartate aminotransferase (AST) level, alanine aminotransferase (ALT) level, C-reactive protein (CRP) level, and model for end-stage liver disease (MELD) score were higher in patients with LC in the post-ACLF group compared with those in the non- and pre-ACLF groups. The PT level was lower in patients with LC in the post-ACLF group compared with those in the non- and pre-ACLF groups.

### 3.2. ADAMTS13:AC and VWF:Ag Levels

The ADAMTS13:AC levels in patients with LC decreased gradually in the order of the non-ACLF group, pre-ACLF group, and post-ACLF group (*p* < 0.05) (Figure 2a). The VWF:Ag levels in patients with LC increased gradually in the order of the non-ACLF group, pre-ACLF group, and post-ACLF group (*p* < 0.05) (Figure 2b). Finally, the VWF:Ag/ADAMTS13:AC levels in patients with LC increased gradually in the order of the non-ACLF group, pre-ACLF group, and post-ACLF group (*p* < 0.05) (Figure 2c).

### 3.3. Relationship between ADAMTS13:AC or VWF:Ag and Other Parameters

ADAMTS13:AC was directly correlated with Alb (*r* = 0.492, *p* < 0.05), PT (*r* = 0.374, *p* < 0.05), and platelet count (*r* = 0.288, *p* < 0.05) (Figure 3a–c). In turn, ADAMTS13:AC was inversely correlated with T-Bil (*r* = −0.492, *p* < 0.05) and Cre (*r* = −0.299, *p* < 0.05) (Figure 3d,e). VWF:Ag was inversely correlated with Alb (*r* = −0.605, *p* < 0.05) and PT (*r* = −0.357, *p* < 0.05) (Figure 3f,g). Conversely, VWF:Ag was directly correlated with T-Bil (*r* = 0.441, *p* < 0.05) and Cre (*r* = 0.236, *p* < 0.05) (Figure 3h,i).

### 3.4. Prognostic Factor for Patients with Liver Cirrhosis in the Post-ACLF Group

The characteristics of the outcomes in patients with LC in the post-ACLF group are shown in Table 2. The age of the survival subgroup in the post-ACLF group was younger than that of the mortality subgroup (*p* < 0.05). The PT and ADAMTS13:AC levels in the survival subgroup of the post-ACLF group were higher than those of the mortality subgroup (*p* < 0.05 for both). The chronic liver failure consortium (CLIF-C) ACLF score, MELD score, VWF:Ag level, and VWF:Ag/ADAMTS13:AC level in the survival subgroup of the post-ACLF group were lower than those of the mortality subgroup (*p* < 0.05 for all). We performed univariate analysis using the factors that were reported previously for the association of prognosis of ACLF. To determine prognostic factors for patients with liver cirrhosis in the post-ACLF group, we performed multivariate analysis using VWF:Ag/ADAMTS13:AC, CLIF-C ACLF score, Cre, and MELD score, which had *p*-values <0.1 in the univariate analysis. The VWF:Ag/ADAMTS13:AC and CLIF-C ACLF scores were associated with prognosis in the post-ACLF group (*p* < 0.05 for both) (Table 3). The time-dependent receiver operating characteristic curve was plotted using VWF:Ag/ADAMTS13:AC and CLIF-C ACLF scores (Figure 4a). The AUC of VWF:Ag/ADAMTS13:AC was 0.823, 0.846, and 0.940 for 30, 60, and 90 days, respectively (Figure 4b–d). The AUC of CLIF-C ACLF score was 0.917, 0.949, and 0.726 for 30, 60, and 90 days, respectively (Figure 4e–g). The cumulative survival rates in the post-ACLF group in patients with low (≤9) and high (>9) VWF:Ag/ADAMTS13:AC are shown in Figure 5. The cumulative survival in the post-ACLF group was significantly lower in patients with a high VWF:Ag/ADAMTS13:AC compared with those with a low VWF:Ag/ADAMTS13:AC (HR: 10.72; 95% confidence interval: 1.39–82.78, *p* < 0.05).

## 4. Discussion

In this study, ADAMTS13:AC, VWF:Ag, and VWF:Ag/ADAMTS13:AC levels gradually decreased, increased, and increased, respectively, in the order of the non-ACLF group, pre-ACLF group, and post-ACLF group. Our previous study reported that the VWF:Ag/ADAMTS13:AC became a predictive biomarker for ACLF development [13]. It is well known that the triggers of ACLF development include bacterial infection, gastrointestinal bleeding, alcohol intake, or worsening of the underlying liver disease. A previous study reported that low ADAMTS13:AC and high VWF:Ag levels are risk factors for spontaneous bacterial peritonitis [2,30], esophagogastric varices [31], and variceal bleeding [30] in patients with LC. Moreover, our previous study revealed that ADAMTS13:AC, VWF:Ag, and VWF:Ag/ADAMTS13:AC levels gradually decreased, increased, and increased, respectively, according to the severity of LC [2,8,9], i.e., ADAMTS13:AC, VWF:Ag, and VWF:Ag/ADAMTS13:AC were associated with functional hepatic reserve, and alcohol intake decreased ADAMTS13:AC levels and increased VWF:Ag and VWF:Ag/ADAMTS13:AC levels in healthy volunteers [13]. As a result, the VWF:Ag/ADAMTS13:AC may be associated with the triggers of ACLF development.

The VWF:Ag/ADAMTS13:AC was demonstrated to be a prognostic biomarker for patients with LC with ACLF in the present study. A previous study reported that ACLF patients with high VWF:Ag levels had a lower survival rate than those with low levels [32]. In addition, ADAMTS13:AC, VWF:Ag, and the VWF:Ag/ADAMTS13:AC levels were decreased, increased, and increased, respectively, according to LC progression [2,8,9,12], and high VWF:Ag and low ADAMTS13:AC levels were prognostic factors in patients with LC [2,14]. Furthermore, it is well known that kidney function is associated with prognosis in ACLF patients [33]. The present study showed that ADAMTS13:AC was inversely correlated and VWF:Ag was directly correlated with Cre. Our previous study reported that patients with low ADAMTS13:AC levels had a higher complication risk of HRS than those with high levels [2,8,9]. The VWF:Ag/ADAMTS13:AC may be associated with the prognosis of ACLF because it is correlated with functional hepatic reserve and kidney function.

Patients with LC have gut dysbiosis [34], which is associated with LC progression and ACLF development [16,17] via an endotoxin (Et) produced by gut microbiota [12,35]. Our previous study reported that Et levels in patients with LC increased according to the severity of LC [12,36], and a high Et level is a predictive biomarker of the development of ACLF in patients with LC [35], and patients with LC and high Et levels had lower ADAMTS13:AC and higher VWF:Ag levels than those with low Et levels [12]. Moreover, a previous study reported that healthy volunteers with intravenous infusion of Et had lower ADAMTS13:AC and higher VWF:Ag levels than those without the infusion [37]. As a result, ADAMTS13:AC, VWF:Ag, and Et were interrelated in LC and ACLF. Additionally, it is well known that Et is associated with sepsis and disseminated intravascular coagulation (DIC). A previous study reported that patients with high VWF:Ag/ADAMTS13:AC levels had a high risk of sepsis with MOF [8,38]. Our previous study reported that VWF:Ag/ADAMTS13:AC was associated with the severity of acute cholangitis and DIC score [1]. Infections, including sepsis, are the most important precipitating events for the development of ACLF. The present study showed that precipitating events for the development of ACLF were 32%. The VWF:Ag/ADAMTS13:AC may be associated with the prognosis of ACLF because it is associated with the severity of infections, as well as functional hepatic reserve and kidney function.

This study had several limitations, including enrolment at a single study center and a small sample size. Moreover, the occasional occurrence of VTE (e.g., portal thrombosis) in patients with LC may affect the VWF:Ag/ADAMTS13:AC [1,39,40]. A previous study reported that the VWF:Ag/ADAMTS13:AC in patients with severe alcoholic hepatitis was recovered in the recovery stage and worsened in the end stage compared with patients at the start of the treatment [11]. Furthermore, the present study did not investigate the change in the VWF:Ag/ADAMTS13:AC after the start of treatment. Therefore, further studies are necessary to investigate this issue.

In summary, the VWF:Ag/ADAMTS13:AC was increased according to progression of ACLF and was independently associated with prognosis in patients with ACLF.

## Figures and Tables

**Figure 1 biology-12-00164-f001:**
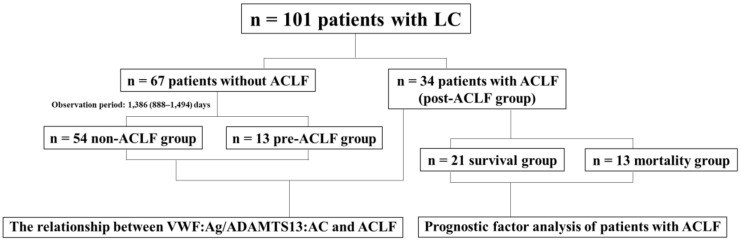
**Study Design.** We initially investigated the relationship between ADAMTS13:AC and VWF:Ag and the non-ACLF group, pre-ACLF group, or post-ACLF group. Subsequently, we assessed the relationship between ADAMTS13:AC and VWF:Ag and prognosis in patients with LC in the post-ACLF group. ACLF, acute-on-chronic liver failure; LC, liver cirrhosis; ADAMTS13, a disintegrin-like and metalloproteinase with thrombospondin type 1 motifs 13; ADAMTS13:AC, ADAMTS13 activity; VWF, von Willebrand factor; VWF:Ag, VWF antigen.

**Figure 2 biology-12-00164-f002:**
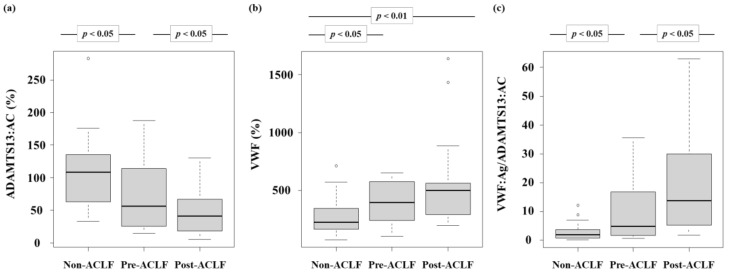
**Plasma ADAMTS13:AC and VWF:Ag levels.** (**a**) ADAMTS13:AC levels in patients with LC gradually decreased in the order of the non-ACLF group, pre-ACLF group, and post-ACLF group (*p* < 0.05). (**b**) VWF:Ag levels in patients with LC gradually increased in the order of the non-ACLF group, pre-ACLF group, and post-ACLF group (*p* < 0.05). (**c**) The VWF:Ag/ADAMTS13:AC levels in patients with LC gradually increased in the order of the non-ACLF group, pre-ACLF group, and post-ACLF group (*p* < 0.05). ACLF, acute-on-chronic liver failure; LC, liver cirrhosis; ADAMTS13, a disintegrin-like and metalloproteinase with thrombospondin type 1 motifs 13; ADAMTS13:AC, ADAMTS13 activity; VWF, von Willebrand factor; VWF:Ag, VWF antigen; VWF:Ag/ADAMTS13:AC, the ratio of VWF:Ag to ADAMTS13:AC.

**Figure 3 biology-12-00164-f003:**
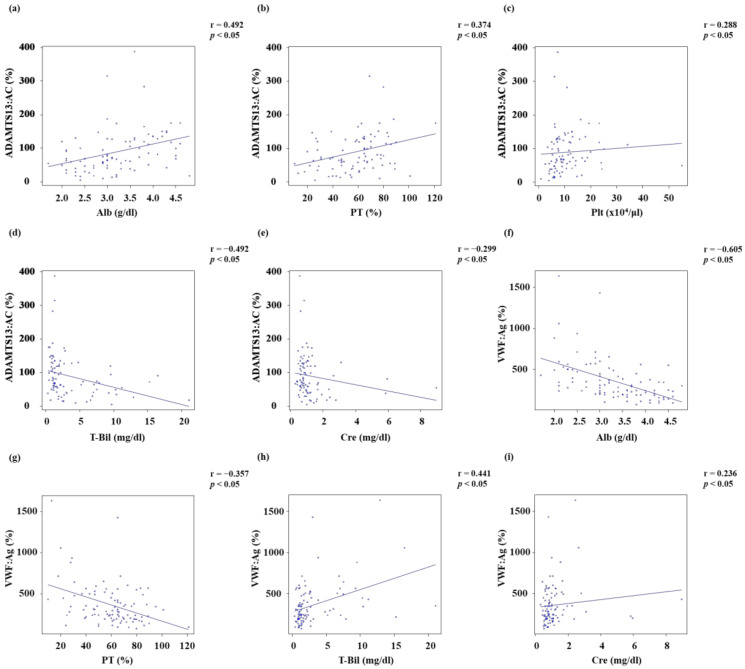
**Relationship between ADAMTS13:AC or VWF:Ag and other parameters.** (**a**–**c**) ADAMTS13:AC was directly correlated with Alb (*r* = 0.492, *p* < 0.05), PT (*r* = 0.374, *p* < 0.05), and Plt (*r* = 0.288, *p* < 0.05). (**d**,**e**) ADAMTS13:AC was inversely correlated with T-Bil (*r* = −0.492, *p* < 0.05) and Cre (*r* = −0.299, *p* < 0.05). (**f**,**g**) VWF:Ag was inversely correlated with Alb (*r* = −0.605, *p* < 0.05) and PT (*r* = −0.357, *p* < 0.05). (**h**,**i**) VWF:Ag was directly correlated with T-Bil (*r* = 0.441, *p* < 0.05) and Cre (*r* = 0.236, *p* < 0.05). Alb, albumin; PT, Prothrombin time; Plt, Platelet count; T-Bil, Total bilirubin; Cre, Creatinine; ADAMTS13, a disintegrin-like and metalloproteinase with thrombospondin type 1 motifs 13; ADAMTS13:AC, ADAMTS13 activity; VWF, von Willebrand factor; VWF:Ag, VWF antigen.

**Figure 4 biology-12-00164-f004:**
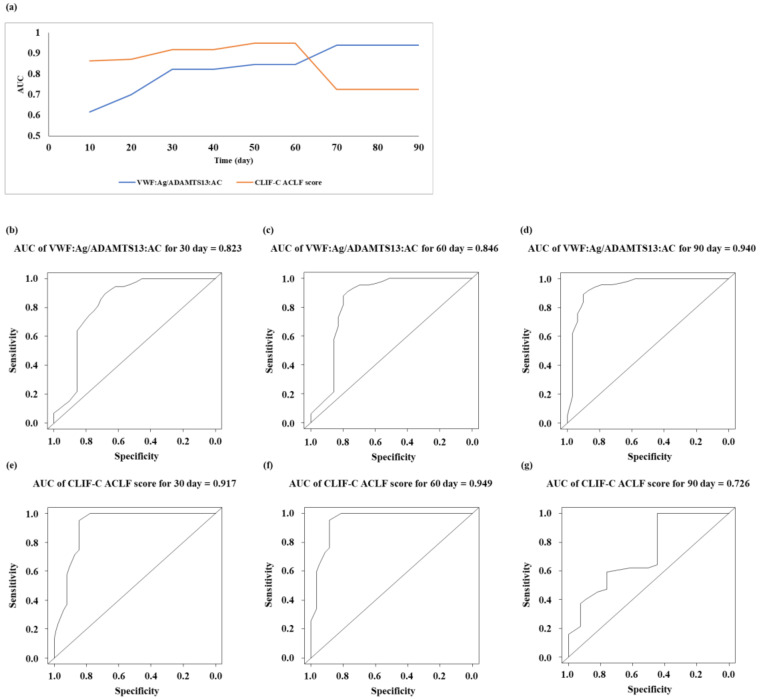
**ROC curve for predicting prognosis in the post-ACLF group by VWF:Ag/ADAMTS13:AC and CLIF-C ACLF score.** (**a**) The time-dependent AUCs of VWF:Ag/ADAMTS13:AC and CLIF-C ACLF score according to days of follow-up. (**b**–**d**) The ROC curve for predicting prognosis in the post-ACLF group by VWF:Ag/ADAMTS13:AC at 30, 60, and 90 days of follow-up; the AUCs of ROC curves are 0.823, 0.846, and 0.940, respectively. (**e**–**g**) The ROC curve for predicting prognosis in the post-ACLF group by CLIF-C ACLF at 30, 60, and 90 days of follow-up; the AUCs of ROC curves are 0.917, 0.949, and 0.726, respectively. ROC, receiver operating characteristic; AUC, area under curve; CLIF-C, the chronic liver failure consortium; ACLF, acute-on-chronic liver failure; ADAMTS13, a disintegrin-like and metalloproteinase with thrombospondin type 1 motifs 13; ADAMTS13:AC, ADAMTS13 activity; VWF, von Willebrand factor; VWF:Ag, VWF antigen; VWF:Ag/ADAMTS13:AC, the ratio of VWF:Ag to ADAMTS13:AC.

**Figure 5 biology-12-00164-f005:**
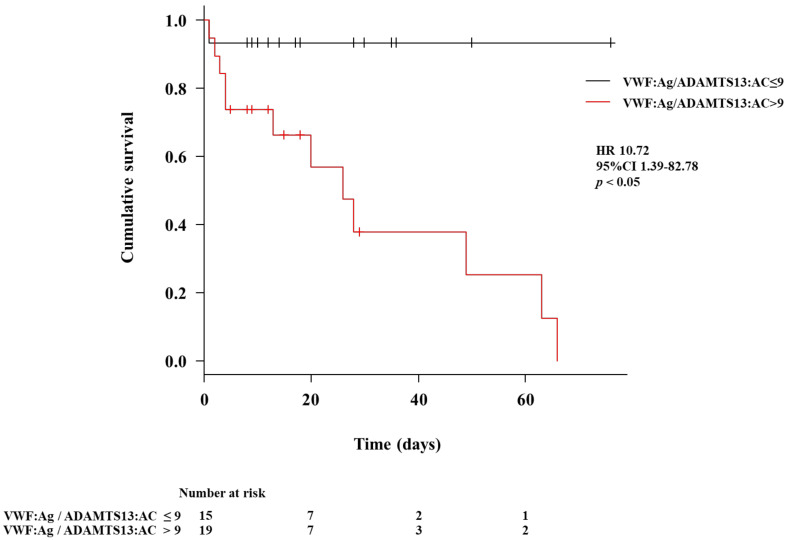
**Cumulative survival according to the VWF:Ag/ADAMTS13:AC in patients with liver cirrhosis in the post-ACLF group.** The patients with LC in the post-ACLF group were categorized into two groups according to the ROC cutoff of the VWF:Ag/ADAMTS13:AC levels (low, ≤9; and high, >9). The cumulative survival was significantly lower in patients with LC in the post-ACLF group with high VWF:Ag/ADAMTS13:AC levels compared with those with low VWF:Ag/ADAMTS13:AC levels (HR, 10.72; 95% CI, 1.39–82.78; *p* < 0.05). ACLF, acute-on-chronic liver failure; ROC, receiver operating characteristic; LC, liver cirrhosis; ADAMTS13, a disintegrin-like and metalloproteinase with thrombospondin type 1 motifs 13; ADAMTS13:AC, ADAMTS13 activity; VWF, von Willebrand factor; VWF:Ag, VWF antigen; VWF:Ag/ADAMTS13:AC, the ratio of VWF:Ag to ADAMTS13:AC; HR, hazard ratio; CI, confidence interval.

**Table 1 biology-12-00164-t001:** Characteristics of patients with liver cirrhosis.

Variable	Total (n = 101)	Non-ACLF Group (n = 54)	Pre-ACLF Group (n = 13)	Post-ACLF Group (n = 34)
Age (years)	71 (62–75)	69 (63–75)	73 (71–81)	71 (62–77)
Sex (male/female)	64/37	32/22	8/5	24/10
Etiology (HBV/HCV/alcohol/NASH/AIH/PBC/others)	5/23/40/14/7/5/7	2/14/18/11/3/4/2	1/5/5/1/0/1/0	2/4/17/2/4/0/5
Albumin (g/dL)	3.2 (2.8–3.9)	3.8 (3.4–4.2)	3.0 (2.9–3.2) *	2.5 (2.1–3.0) *^,^ **
Aspartate aminotransferase (U/L)	43 (32–76)	36 (28–50)	42 (39–69)	80 (42–232) *^,^ **
Alanine aminotransferase (U/L)	28 (19–58)	24 (17–41)	27 (23–39)	65 (30–113) *^,^ **
Blood urea nitrogen (mg/dL)	16 (13–25)	15 (12–21)	16 (14–20)	22 (12–37) *
Creatinine (mg/dL)	0.9 (0.7–1.2)	0.8 (0.5–1.0)	1.0 (0.9–1.1) *	1.1 (0.7–1.7) *
Total bilirubin (mg/dL)	1.6 (1.0–4.1)	1.1 (0.9–1.6)	1.8 (1.1–2.2) *	7.1 (5.0–10.2) *^,^ **
Prothrombin time (%)	61 (40–73)	65 (55–73)	67 (62–79)	33 (28–40) *^,^ **
Platelet count (×10^4^/mm^3^)	9.2 (6.3–12.8)	9.6 (7.5–12.8)	6.8 (5.5–13.0)	7.7 (6.0–11.8) *
C-reactive protein (mg/dL)	0.3 (0.0–1.0)	0.1 (0.0–0.3)	0.2 (0.1–0.6)	1.5 (0.9–4.3) *^,^ **
MELD score	10 (9–17)	9 (8–11)	9 (7–12)	20 (14–28) *^,^ **

Data are expressed as median (interquartile range). HBV, hepatitis B virus; HCV, hepatitis C virus; NASH, non-alcoholic steatohepatitis; AIH, autoimmune hepatitis; PBC, primary biliary cholangitis; MELD, model for end-stage liver disease. * *p* < 0.05 between non-ACLF and pre-ACLF, or post-ACLF group. ** *p* < 0.05 between pre-ACLF and post-ACLF group.

**Table 2 biology-12-00164-t002:** Characteristics of patients in the post-ACLF group.

Variable	Total (n = 34)	Survival Group (n = 21)	Mortality Group (n = 13)	*p*-Value *
Age (years)	71 (62–77)	63 (47–73)	77 (65–81)	0.022
Sex (male/female)	24/10	14/7	10/3	0.704
Etiology (HBV/HCV/alcohol/NASH/AIH/others)	2/4/17/2/4/5	0/1/13/1/3/3	2/3/4/1/1/2	0.155
Events (infection/bleeding/alcohol/underlying liver disease/circulatory insufficiency/others)	11/5/8/7/1/2	6/3/6/5/0/1	5/2/2/2/1/1	0.794
Severity grade of ACLF (0/1/2/3)	20/2/4/8	18/2/1/0	2/0/3/8	0.000
CLIF-C ACLF score	44 (38–52)	40 (37–45)	52 (45–58)	0.002
Albumin (g/dL)	2.5 (2.1–3.0)	2.5 (2.3–3.0)	2.5 (2.1–2.9)	0.301
Aspartate aminotransferase (U/L)	80 (42–232)	74 (47–226)	71 (43–240)	0.808
Alanine aminotransferase(U/L)	62 (30–113)	57 (29–90)	61 (22–133)	0.955
Blood urea nitrogen (mg/dL)	22 (12–37)	26 (12–34)	22 (21–41)	0.600
Creatinine (mg/dL)	1.1 (0.7–1.7)	1.2 (0.7–1.5)	1.2 (1.0–1.9)	0.203
Total bilirubin (mg/dL)	7.1 (5.0–10.2)	6.0 (5.0–7.5)	9.7 (6.1–10.7)	0.062
Prothrombin time (%)	33 (28–40)	38 (30–40)	28 (22–30)	0.016
Platelet count (×10^4^/mm^3^)	7.7 (6.0–11.8)	8.0 (6.4–13.1)	8.2 (5.9–10.9)	0.779
C-reactive protein (mg/dL)	1.5 (0.9–4.3)	1.2 (0.7–3.9)	2.0 (1.0–4.5)	0.339
MELD score	20 (14–28)	18 (13–25)	25 (22–30)	0.018
ADAMTS13:AC (%)	41 (18–67)	63 (29–71)	35 (14–44)	0.038
VWF:Ag (%)	502 (298–763)	314 (242–561)	819 (444–1560)	0.007
VWF:Ag/ADAMTS13:AC	13.7 (6.1–29.5)	7.4 (3.7–14.1)	19.6 (9.8–31.2)	0.009

Data are expressed as median (interquartile range). *p*-values represent comparisons of patients in the post-ACLF group between the survival and mortality subgroups. HBV, hepatitis B virus; HCV, hepatitis C virus; NASH, non-alcoholic steatohepatitis; AIH, autoimmune hepatitis; CLIF-C, the chronic liver failure consortium; ACLF, acute-on-chronic liver failure; MELD, model for end-stage liver disease; ADAMTS13, a disintegrin-like and metalloproteinase with thrombospondin type 1 motifs 13; ADAMTS13:AC, ADAMTS13 activity; VWF, von Willebrand factor; VWF:Ag, VWF antigen; VWF:Ag/ADAMTS13:AC, the ratio of VWF:Ag to ADAMTS13:AC. * Survival group vs. mortality group.

**Table 3 biology-12-00164-t003:** The association of patient characteristics and outcomes.

Variable	Univariate Analysis			Multivariate Analysis		
	HR	95%CI	*p*-Value	HR	95%CI	*p*-Value
ADAMTS13:AC	0.99	0.950–1.023	0.4409			
VWF:Ag	1.00	1.000–1.002	0.2601			
VWF:Ag/ADAMTS13:AC	1.01	0.999–1.012	0.0789	1.01	1.001–1.009	0.0261
CLIF-C ACLF score	1.17	1.078–1.273	0.0002	1.19	1.039–1.404	0.0138
Creatinine	1.63	1.152–2.296	0.0057	1.20	0.610–2.401	0.5844
C-reactive protein	1.04	0.925–1.176	0.4956			
MELD score	1.10	1.039–1.172	0.0013	0.96	0.803–1.156	0.6885

The analysis was evaluated using Cox proportional hazards regression analysis. ADAMTS13, a disintegrin-like and metalloproteinase with thrombospondin type 1 motifs 13; ADAMTS13:AC, ADAMTS13 activity; VWF, von Willebrand factor; VWF:Ag, VWF antigen; VWF:Ag/ADAMTS13:AC, the ratio of VWF:Ag to ADAMTS13:AC; CLIF-C, the chronic liver failure consortium; ACLF, acute-on-chronic liver failure; MELD, model for end-stage liver disease; HR, hazard ratio; CI, confidence interval.

## Data Availability

The data are not publicly available due to privacy.

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
