# Peer review of "The Ratio of von Willebrand Factor Antigen to ADAMTS13 Activity: Usefulness as a Prognostic Biomarker in Acute-on-Chronic Liver Failure"

_biology, 2023, doi:10.3390/biology12020164_

Round 1

Reviewer 1 Report

Takuya et al present a study investigating the prognostic value of VWF:Ag/ADAMTS13:AC in patients with ACLF, as well as differences in VWF:Ag/ADAMTS13:AC levels between patient with ACLF, patients, who develop ACLF and patients without ACLF. This is interesting and novel, however there are a number of important issues that need to be addressed.

1) The study investigates the data of patients with LC, who are hospitalized. However, it is not indicated, why these patients are hospitalized, which is very important. How many of them had infections and how many had portal vein thrombosis (which are known to substantially influence VWF antigen)?

2) The authors use the term 'uncontrolled hepatocellular carcinoma'. It is unclear, what they mean by this and this should be clarified.

3) The lowest possible MELD score is 6 points. How can it be, that the IQR of the MELD in Table 1 in the Before ACLF-development group ranges from 4-8?

4) Based on the information provided by the authors in the manuscript, it is unclear, which benefit the assessment of VWF:Ag/ADAMTS13:AC offers over VWF:Ag alone. 

5) Lines 296-305: These hypotheses are highly speculative and not backed by the data. Based on the data provided, no statements about possible microthrombi generation or things like that can be taken. While ADAMTS13:AC may be decreased in patients with thrombotic thrombocytopenia purpura, this cannot be just applied to patients with liver cirrhosis, since cirrhosis is a completely unique entity when it comes to hemostasis with a complex dysregulation of various pro- and antithrombotic factors.

6) Line 220-230: Again, the authors speculate about the pathophysiology of ACLF without support by the data they provide. They did not study micro thrombi, nor did they study endotoxin. Thus, they should not spend a major part of the discussion hypothesizing about these factors and their potential implications in ACLF development. Rather, they should focus on discussing the results that they are actually presenting.

7) The authors should modify the last sentence since they did not show that their parameter is associated with "the pathophysiology of ACLF".

Minor comments:

1) the naming of the groups is confusing. In my opinion, the groups of patients with ACLF should simply be called 'ACLF pre-baseline group' and 'ACLF post-baseline group'.

Author Response

Response to the Reviewer 1

Thank you for reviewing our manuscript. Our responses to your comments are as follows:

1)The study investigates the data of patients with LC, who are hospitalized. However, it is not indicated, why these patients are hospitalized, which is very important. How many of them had infections and how many had portal vein thrombosis (which are known to substantially influence VWF antigen)?

Response to 1)

Thank you for your valuable comment. We have indicated that the non- and pre-ACLF groups had samples collected during hospital visits and the post-ACLF group had samples collected on admission of the patient. We have also mentioned the number of patients with infections and venous thromboembolism. We have revised the patients and study design subsection of the Methods section as follows:

; 34 patients with ACLF did not have venous thromboembolism (VTE). Samples were collected on admission for ACLF. A total of 11 of 34 patients with ACLF had infections (4 had cellulitis, 3 had sepsis, 2 had spontaneous bacterial peritonitis, 1 had appendicitis, and 1 had pyelonephritis). Sixty-seven patients without ACLF did not have VTE or infection. Samples were collected during hospital visits for LC.

2)The authors use the term 'uncontrolled hepatocellular carcinoma'. It is unclear, what they mean by this and this should be clarified.

Response to 2)

Thank you for your valuable comment. We changed “uncontrolled” to “incurable.”

3)The lowest possible MELD score is 6 points. How can it be, that the IQR of the MELD in Table 1 in the Before ACLF-development group ranges from 4-8?

Response to 3)

We apologize for miswriting the MELD score values. Table 1 show the revised MELD score values.

4)Based on the information provided by the authors in the manuscript, it is unclear, which benefit the assessment of VWF:Ag/ADAMTS13:AC offers over VWF:Ag alone.

Response to 4)

Thank you for your insightful comment. Table 3 shows that VWF:Ag was not associated with prognosis in the post-ACLF group.

5)Lines 296-305: These hypotheses are highly speculative and not backed by the data. Based on the data provided, no statements about possible microthrombi generation or things like that can be taken. While ADAMTS13:AC may be decreased in patients with thrombotic thrombocytopenia purpura, this cannot be just applied to patients with liver cirrhosis, since cirrhosis is a completely unique entity when it comes to hemostasis with a complex dysregulation of various pro- and antithrombotic factors.

6)Line 220-230: Again, the authors speculate about the pathophysiology of ACLF without support by the data they provide. They did not study micro thrombi, nor did they study endotoxin. Thus, they should not spend a major part of the discussion hypothesizing about these factors and their potential implications in ACLF development. Rather, they should focus on discussing the results that they are actually presenting.

Response to 5,6

Thank you for your crucial comment. We have removed the noted sentence and added the association between VWF:Ag/ADAMTS13 and infection. We have revised the Discussion section as follows:

As a result, ADAMTS13:AC, VWF:Ag, and Et were interrelated in LC and ACLF. Additionally, it is well-known that Et is associated with sepsis and disseminated intravascular coagulation (DIC). A previous study reported that patients with high VWF:Ag/ADAMTS13:AC levels had a high risk of sepsis with MOF. Our previous study reported that VWF:Ag/ADAMTS13:AC was associated with the severity of acute cholangitis and DIC score. Infections, including sepsis, are the most important precipitating events for the development of ACLF. The present study showed that the precipitating events for the development of ACLF was 32%. The VWF:Ag/ADAMTS13:AC may be associated with the prognosis of ACLF because it is associated with the severity of infections, as well as functional hepatic reserve and kidney function.

7)The authors should modify the last sentence since they did not show that their parameter is associated with "the pathophysiology of ACLF".

Response to 7)

Thank you for your valuable comment. We changed “the VWF:Ag/ADAMTS13:AC was associated with the pathophysiology of ACLF” to “the VWF:Ag/ADAMTS13:AC was increased according to progression of ACLF” in the Discussion section.

8)The naming of the groups is confusing. In my opinion, the groups of patients with ACLF should simply be called 'ACLF pre-baseline group' and 'ACLF post-baseline group'.

Response to 8)

Thank you for your valuable comment. Accordingly, we have changed “before-ACLF development group” to “pre-ACLF group” and “after-ACLF development group” to “post-ACLF group.”

Reviewer 2 Report

Major points   Information on compensated vs. decompensated stage of the patients is missing. The MELD of the cohort is quite low - how was the presence of liver cirrhosis ascertained?   A supplementary analysis applying the EF-CLIF criteria for ACLF should be performed.   Exact p-values should be provided.   Dichotomization of the VWF/ADAMTS13 ratio is questionable. See, e.g., https://www.bmj.com/content/332/7549/1080.1Determination of the "ideal" cut-off by Youden's index and comparison with arbitrarily chosen cut-offs for other variables is not an ideal approach. Start with a multivariable analyses of continuous variables. Comparison of HRs between dichotomized variables is not informative - I would suggest du perform time-dependent AUROC analyses for VWF, ADAMTS13, the ratio, and well-established prognostic indicators such as MELD/CRP.   Severity of systemic inflammation is a well-established prognostic indicator and may impact both VWF and ADAMTS13 - please provide information and adjust your analysis accordingly.

Author Response

Response to the Reviewer 2

Thank you for reviewing our manuscript. Our responses to your comments are as follows:

1)Information on compensated vs. decompensated stage of the patients is missing. The MELD of the cohort is quite low - how was the presence of liver cirrhosis ascertained? A supplementary analysis applying the EF-CLIF criteria for ACLF should be performed. Exact p-values should be provided.

Response to 1)

Thank you for your valuable comment. We apologize for miswriting MELD score values. Table 1 show the revised MELD score values, and Table 2 shows the severity grade of ACLF. In this study, we have used the diagnostic criteria for acute-on-chronic liver failure and related disease conditions in Japan. In addition, Tables 2 and 3 show exact p-values.

2)Dichotomization of the VWF/ADAMTS13 ratio is questionable. See, e.g., https://www.bmj.com/content/332/7549/1080.1Determination of the "ideal" cut-off by Youden's index and comparison with arbitrarily chosen cut-offs for other variables is not an ideal approach. Start with a multivariable analyses of continuous variables. Comparison of HRs between dichotomized variables is not informative - I would suggest du perform time-dependent AUROC analyses for VWF, ADAMTS13, the ratio, and well-established prognostic indicators such as MELD/CRP. Severity of systemic inflammation is a well-established prognostic indicator and may impact both VWF and ADAMTS13 - please provide information and adjust your analysis accordingly.

Response to 2)

Thank you for your pertinent comment. In this study, the statistical analysis was advised by Takashi Inoue (included in the acknowledgements section), who specializes in statistics. We found that multivariable analysis was difficult because the sample size was small. In addition, we used EZR for the statistical analysis. Unfortunately, EZR cannot be performed on time-dependent AUROC analyses.

3)Severity of systemic inflammation is a well-established prognostic indicator and may impact both VWF and ADAMTS13 - please provide information and adjust your analysis accordingly.

Response to 3)

Thank you for your valuable comment. We have interpreted severity of SIRS as severity of ACLF. Tables 2 and 3 show the severity grade of ACLF.

Round 2

Reviewer 1 Report

Thank you for the answers to my comment. All my comments have been satisfactorily answered, in my opinion.

Author Response

Thank you for reviewing our manuscript.

Reviewer 2 Report

Unfortunately, my comments number 1 and 2 have not been adequately addressed. Data on ACLF by EF-CLIF is essential to evaluate the generalizibility of the findings. Fully relying on dichotomized data is unacceptable and there are also other statistical softwares that can be used.

Author Response

Thank you for reviewing our manuscript. Our responses to your comments are as follows:

Data on ACLF by EF-CLIF is essential to evaluate the generalizibility of the findings. Fully relying on dichotomized data is unacceptable and there are also other statistical softwares that can be used.

Response

Thank you for your valuable comment. We have performed multivariate analysis using VWF:Ag/ADAMTS13:AC, CLIF-C ACLF score, and other factors of continuous variables. After that, we have indicated the time-dependent receiver operating characteristic curve using the VWF:Ag/ADAMTS13:AC and CLIF-C ACLF score. We have revised the prognostic factor for patients with liver cirrhosis in the post-ACLF group in the Result section as follows:

We performed univariate analysis using the factors that were reported previously the association of prognosis of ACLF . To determine prognostic factors for patients with liver cirrhosis in the post-ACLF group, we performed multivariate analysis using VWF:Ag/ADAMTS13:AC, CLIF-C ACLF score, Cre, and MELD score, which had P-values <0.1 in the univariate analysis. The VWF:Ag/ADAMTS13:AC and CLIF-C ACLF scores were associated with prognosis in the post-ACLF group (P < 0.05 for both) (Table 3). The time-dependent receiver operating characteristic curve was plotted using VWF:Ag/ADAMTS13:AC and CLIF-C ACLF scores (Fig. 4a). The AUC of VWF:Ag/ADAMTS13:AC was 0.823, 0.846, and 0.940 for 30, 60, and 90 days, respectively (Fig. 4b, c and d). The AUC of CLIF-C ACLF score was 0.917, 0.949, and 0.726 for 30, 60, and 90 days, respectively (Figure 4e, f and g).

Round 3

Reviewer 2 Report

The authors have addressed my comments sufficiently.